# Determining the minimal important differences in the International Prostate Symptom Score and Overactive Bladder Questionnaire: results from an observational cohort study in Dutch primary care

Marco H Blanker [1], Harma Johanna Alma,[1,2] Tahira Sakina Devji [3], Marjan Roelofs,[1] Martijn G Steffens,[4] Henk van der Worp [1]

For numbered affiliations see end of article.

**Correspondence to**
Dr Marco H Blanker;
m.h.blanker@umcg.nl

## ABSTRACT

**Objectives** To determine the minimal important difference (MID) of the International Prostate Symptom Score (IPSS) and the Overactive Bladder Questionnaire short form (OAB-q SF) assessed in primary care among patients treated for lower urinary tract symptoms (LUTS).

**Design** Single-arm, open-label observational cohort study with a 6-week follow-up.

**Setting** Twenty-two pharmacies in the Netherlands.

**Participants** We enrolled Dutch men with uncomplicated LUTS who received a new alpha-blocker prescription from their general practitioner or urologist.

**Primary and secondary outcomes** The IPSS and OAB-q SF were completed before and after 6 weeks of therapy. At 6 weeks, men also completed the Patient Global Impression of Improvement (PGI-I). The mean change scores of the IPSS and OAB-q SF were calculated for each PGI-I outcome category, with the category 'a little better' used to determine the MID. The SE of measurement (SEM) was calculated for each questionnaire.

**Results** In total, 165 men completed follow-up. The MID was 5.2 points (95% CI 3.9 to 6.4; SEM 3.6) for the IPSS and 11.0 points (95% CI 7.1 to 14.9; SEM 9.7) for the OAB-q SF. For both questionnaires, CIs showed an overlap with the no-change categories. However, the MID for the IPSS was higher in men with severe baseline symptoms (7.1; 95% CI 5.3 to 9.0) than in men with moderate baseline symptoms (3.2; 95% CI 1.7 to 4.8).

**Conclusion** In this study, the MID for the IPSS was considerably higher than the MID of 3.1 reported in the only other study on this topic, but may be due to methodological differences. Interpretation of the MID for the OAB-q SF is hampered by the overlap with the SEM. Future studies are needed to confirm our results because correlations between the PGI-I and symptom questionnaires were suboptimal.

## INTRODUCTION

Symptom severity is a key outcome for patients with lower urinary tract symptoms (LUTS) and is most often evaluated by direct patient inquiry, using patient-reported outcome measures (PROMs). Although the International Prostate Symptom Score (IPSS) is most often used for this purpose in both clinical trials and practice,[1–5] it fails to capture problematic symptoms such as urinary incontinence and urgency. Therefore, the Overactive Bladder Questionnaire (OAB-q) is increasingly being used to evaluate the treatment of OAB,[6 7] with the short form (ie, OAB-q SF) having the advantage of being less time consuming.[8] Together, both of the IPSS and OAB-q SF capture the spectrum of outcomes that are important to patients, but it is difficult to interpret the effects of an intervention expressed as mean scores or change scores over time.

The minimal important difference (MID) has proven invaluable when interpreting

PROMs and could be of great value for both the IPSS and OAB-q SF.[9 10] To date, the MID has only been reported for the IPSS in a study conducted in secondary care among participants of clinical trials.[1 2] There has been no report of the MID for the OAB-q SF in any care setting. In countries like the Netherlands and the UK, most men with LUTS first visit their general practitioner(GP) to seek treatment. Given that setting may affect the MID, possibly because of differences in baseline symptom severity,[9–11] we feel that it is important to assess the MID in a primary care setting. To date the MID for secondary care settings has been applied in guidelines for primary care.[3 5] It is unclear if applying the threshold for a clinically relevant outcome is appropriate. Men who receive treatment need to be aware of what can be expected. Knowledge about the MID in primary care will then provide invaluable data for interpreting treatment outcomes that may differ between primary and secondary care. In addition, evidence must be obtained from multiple studies to ensure that MID determinations are accurate.

In the current study, we aimed to determine the MIDs for both the IPSS and OAB-q SF in a patient cohort originating mainly from primary care.

## METHODS
### Study design
We conducted a prospective cohort study between January 2016 and April 2018.[12] Baseline data for the IPSS and OAB-q SF were compared with follow-up data after 6 weeks of treatment. At follow-up, participants also completed the Patient Global Impression of Improvement (PGI-I) and we calculated the MID.

### Participants
Adult men who visited a participating pharmacy in the north of the Netherlands were included if they received a new alpha-blocker prescription for uncomplicated LUTS from a GP or urologist. A prescription was defined as new if no alpha-blocker prescription had been given within the past year. The pharmacists checked if the alpha-blocker was indicated for LUTS and excluded men prescribed alpha-blockers for urinary tract stones or indwelling catheters. All participants provided written informed consent.

### Data collection
At baseline, before starting alpha-blocker therapy, all participants provided relevant descriptive data (eg, age, duration of LUTS in months or years and history of surgery for LUTS) and completed the Dutch versions of the IPSS and OAB-q SF. After 6 weeks, men who consented repeated the IPSS and OAB-q SF by postal invitation. At this time, we asked participants to complete the PGI-I questionnaire.[13] The period of 6 weeks was chosen as clinical effects of alpha-blockers take a few weeks to develop fully, but significant efficacy over placebo can occur within hours to days.[4 14]

### Questionnaires
The IPSS questionnaire was originally validated as the *American Urological Association Symptom Index for benign prostatic hyperplasia*.[1] It includes seven questions covering frequency, nocturia, weak urinary stream, hesitancy, intermittence, incomplete emptying and urgency. Each question has response options ranging from 0 to 5, with higher scores reflecting more severe symptoms. Total scores that may range from 0 (no symptoms) to 35 points (maximum score), and scores are often categorised as no/mild symptoms (0–7 points), moderate symptoms (8–19 points), or severe symptoms (≥20 points). The questionnaire was internally consistent (Cronbach's alpha=0.86) and has excellent test–retest reliability (r=0.92).[1] The MID for the IPSS is currently considered to be 3.1 points.[2] The American Urological Association - Symptom Index (AUA-SI) has been internationally adopted and implemented worldwide under the name IPSS.

IPSS focuses on the concept of 'benign prostatic hyperplasia' as cause of male LUTS, which appeared to have a multifactorial origin. OAB is one of the alternative explanations of LUTS. Although urgency (included in the IPSS) relates to OAB, OAB includes other symptoms as well, which are not included in the IPSS questionnaire. Therefore, Coyne *et al* developed a condition specific questionnaire, the OAB-q.[6 7] The OAB-q was developed from focus groups of men and women, clinician opinion and a thorough literature review. More recently, this OAB-q has been shortened to benefit patients, researchers and clinicians.[8] The OAB-q SF contains six questions on 6-point Likert-type scales, with the outcomes transformed to a 0–100 point scale in which higher scores indicate more severe symptoms.[8] This scale demonstrated good convergent validity, discriminant validity, internal reliability, reproducibility and responsiveness to change.[8]

Both IPSS and OAB-q-SF capture symptoms that are not by definition patient important, but rather reflect the conditions under study. To study if changes on a questionnaire over time are relevant for patients, the PGI-I has been developed using a quantitative approach.[13 15]

The PGI-I is a validated generic tool for assessing overall improvement after treatment and is answered on a 7-point Likert-type scale, with the following options: 'very much better', 'much better', 'a little better', 'no change', 'a little worse', 'much worse' or 'very much worse'.[13 15] Full versions of these questionnaires are presented as online supplementary file 1.

We sent a reminder after 2 weeks to patients who did not respond to follow-up requests.

### Data analyses
Baseline characteristics are reported as continuous variables and summarised as mean and SD or as median and IQR, depending on the distribution checked by the Shapiro-Wilk test. These characteristics were also compared between men with and without completed follow-up data to test for selective non-response. Next, the change scores of the IPSS and OAB-q SF were calculated

by comparing the data between the baseline and follow-up questionnaires. Change scores were inverted to facilitate intuitive interpretation, with positive scores reflecting symptom improvement.

Various methods exist to determine the MID of questionnaires and are typically either anchor-based or distribution-based.[9 16–18] The latter involve evaluating change in the PROM with the probability that the change occurred by chance, sample variation or measurement precision; however, they do not reflect patient perspectives.[17 19] Thus, we used an anchor-based method,[9] in which we compared changes in the IPSS or OAB-q SF (PROM) with the PGI-I (the anchor). For each PGI-I category, we then present the mean change in scores from baseline to follow-up with the associated CIs. We defined the MID as the mean change in IPSS or OAB-q SF for the PGI-I category 'a little better', as the M in MID reflects the *minimal* change that is considered relevant. We also present the mean change scores for the other PGI-I categories.

The usefulness of anchor-based approaches depends on the relationship between the PROM and the anchor.[20–22] The anchor and PROM should be measuring the same or similar underlying constructs and should therefore be appreciably correlated. Correlations between questionnaire change scores and the anchor PGI-I should be obviously strong, as else these measure different concepts. Correlations between the anchor PGI-I and the baseline and follow-up questionnaire scores are performed to check for a possible response shift. Mostly anchor PGI-I scores seem correlated with follow-up scores (due to response shift). We therefore examined the Spearman correlation coefficients between the PGI-I and the IPSS and OAB-q SF for the baseline, follow-up and change data to ensure the anchor's validity. A correlation coefficient between the symptom change scores and the PGI-I of ≥0.50, and an equal and opposite correlation of the PGI-I with the baseline score and the follow-up score, were considered ideal and likely to yield trustworthy MID estimates.[20–22]

To test the impact of baseline symptom severity on the distribution of results, a stratified analysis was performed for the IPSS categories 'moderate symptoms' and 'severe symptoms' because previous research has shown that such stratification has a large impact.[2] No such categories have been defined for the OAB-q SF, so we did not perform a similar analysis for this questionnaire. Subgroup analyses were also performed with participants who received their prescription from their GP, allowing us to provide data that focused on the primary care setting. Finally, we checked if the MID exceeded the measurement error.[15 17] For this, we calculated the SE of measurement (SEM) as follows: $(SD \times (1 - reliability)^{1/2})$. Cronbach's alpha was used as the reliability measure.[23]

The complete data set was used without imputing missing data. All analyses were performed using IBM SPSS V.25.0 (IBM Corp., Armonk, New York, USA), and we considered a p value <0.05 to be statistically significant.

## Patient and public involvement

This study was performed without patient involvement. We did not invite patients to comment on the study design nor did we consult them to interpret the results. Patients were not invited to contribute to the writing or editing of this document for readability or accuracy.

## RESULTS

A total of 251 men completed the baseline questionnaires, of which 165 also completed the follow-up questionnaires. The baseline characteristics of men with and without follow-up data are shown in table 1, with no statistically significant differences found between these groups. Notably, 86.3% of the participants received their prescription from a GP and the remainder received it from a urologist.

There were mean improvements in the IPSS and OAB-q SF scores during the study of 5.8 (SD 6.7) and 11.8 points (SD 17.4), respectively. Between baseline and follow-up at 6 weeks, the mean IPSS changed from 19.1 (SD 6.8) to 13.3 (SD 6.5) and the mean OAB-q SF score changed from 39.7 (SD 19.2) to 27.9 (SD 16.9). The PGI-I outcomes are shown in table 2 and indicate that most men reported that they were 'a little better' or 'much better' (74.7%), while only 23.5% perceived no change. Only three men (1.8%) reported 'worsened' symptoms, and none of the participants reported 'much worsened' or 'very much worsened' symptoms.

Table 2 also shows the distribution of changes in the IPSS and OAB-q SF for each PGI-I category. The MID for the IPSS was 5.2 points (95% CI 3.9 to 6.4) and the PGI-I outcomes 'no change' and 'much better' corresponded to IPSS symptom changes of 3.1 points (95% CI 1.1 to 5.1) and 8.7 points (95% CI 6.8 to 10.7), respectively. The MID for the OAB-q SF was 11.0 points (95% CI 7.1 to 14.9) and the PGI-I outcomes 'no change' and 'much better' corresponded with mean improvements of 3.0 points (95% CI −2.3 to 8.4) and 19.1 points (95% CI 14.3 to 24.0), respectively. For both questionnaires, the CIs of the MID-categories showed an overlap with the 'no change' categories.

The Spearman correlation coefficients were then calculated between the PGI-I and both the IPSS and the OAB-q SF. The correlation was −0.51 for the PGI-I and baseline IPSS, 0.43 for the PGI-I and follow-up IPSS and 0.38 for the PGI-I and change in IPSS. The corresponding correlations for the OAB-q SF were −0.09 at baseline, 0.36 at follow-up and 0.42 for the change.

Subgroup analyses of data for men with a prescription from a GP found no relevant differences, with MID values of 5.4 for the IPSS and 11.2 for the OAB-q SF (table 3). Stratified analysis of baseline data revealed that men with severe symptoms had higher MID values for the IPSS, reaching 7.1 (95% CI 5.3 to 9.0), compared with the MID value of 3.2 (95% CI 1.7 to 4.8) for men with moderate symptoms (table 4).

**Table 1** Baseline characteristics of all participants and participants who dropped out after the baseline questionnaire

| | Participants with completed follow-up (n=165) | Drop out after baseline (n=86) | P value |
|---|---|---|---|
| Age (mean±SD) | 66.7±9.7 | 65.4±12.1 | 0.42* |
| Prescription from GP (%) | 86.3 | 83.2 | 0.55† |
| IPSS (mean±SD) | 19.1±6.8 | 17.6±6.5 | 0.11* |
| IPSS categories (%) | | | 0.12 |
| None/mild | 3.7 | 6.4 | |
| Moderate | 50.9 | 61.5 | |
| Severe | 45.3 | 32.1 | |
| IPSS quality of life (median \| IQR) | 4.0 \| 3.0–5.0 | 4.0 \| 3.0–5.0 | 0.52‡ |
| OAB-q SF (mean±SD) | 39.8±19.2 | 40.7±18.1 | 0.70* |
| Duration of LUTS in months (median \| IQR) | 24.0 \| 5.0–42.0 | 12.0 \| 3.0–36.0 | 0.11‡ |
| History of surgery for LUTS (%) | 1.2 | 3.8 | 0.19† |

*Student's t-test.
†$\chi^2$ test.
‡Mann-Whitney U test.
GP, General practitioner; IPSS, International Prostate Symptom Score; IQR, Interquartile Range; LUTS, lower urinary tract symptoms; OAB-q SF, Overactive Bladder Questionnaire short form.

Finally, the SEM was 3.6 for the IPSS and 9.7 for the OAB-q SF.

## DISCUSSION

We estimated the MID for two questionnaires that are often used to assess male LUTS in primary care. However, whereas the SEM of the IPSS was less than the 95% CI of the MID (5.2 points; 95% CI 3.9 to 6.4; SEM 3.6), the SEM of the OAB-q SF fell within the 95% CI of the MID (11 points; 95% CI 7.1 to 14.9; SEM 9.7). Thus, we can only conclude that the outcomes for the IPSS were unlikely to have occurred because of chance or measurement imprecision. Given that many questionnaires have used multiple MID values, we were surprised to find only one previous estimate of the MID for the IPSS in the literature.[1 2] Our study therefore adds relevant information in the primary care setting for clinicians and guideline developers.

Our results for the IPSS were different to those of the seminal study on this topic performed by Barry et al in secondary care.[1 2] In that study, the MID of 3.1 points (SD 0.27) fell within the 95% CI of the 'no change' group (consisting of men who expressed that they hadn't experienced any change in symptoms), but outside the CI of the 'a little better' group, suggesting a likely underestimation of the real value. In the current study, there was also some overlap between the CIs of the 'no change' and the 'a little better' group, though this was within a change of only 3.9–5.1 points. This could be explained by the relatively small samples in the subgroup analyses. Given that treatment is typically in primary care, we have therefore provided additional data that is applicable to most men with LUTS. Nevertheless, the differences in outcomes compared with the study by Barry et al need to be explained. It is our contention

**Table 2** Change scores for the IPSS and OAB-q SF by PGI-I outcomes

| PGI-I outcome | N (%) | IPSS | Missing | OAB-q SF | Missing |
|---|---|---|---|---|---|
| Very much better | 6 (3.6) | 13.4 (2.9 to 23.9) | 1 | 23.8 (2.3 to 45.3) | 0 |
| Much better | 50 (30.3) | 8.7 (6.8 to 10.7) | 2 | 19.1 (14.3 to 24.0) | 3 |
| A little better | 68 (41.2) | 5.2 (3.9 to 6.4) | 3 | 11.0 (7.1 to 14.9) | 4 |
| No change | 38 (23.0) | 3.1 (1.1 to 5.1) | 0 | 3.0 (−2.3 to 8.4) | 4 |
| A little worse | 3 (1.8) | −5.0 (−30.9 to 20.9) | 0 | −9.7 (−81.7 to 62.4) | 0 |

Change in IPSS and OAB-q SF scores were estimated by comparing symptom scores between baseline and 6 weeks. Mean change and 95% CIs are presented. Outcomes are inverted so that positive changes reflect symptom improvement. The PGI-I category 'a little better' reflects the MID for both questionnaires. None of the participants scored 'much worsened' or 'very much worsened' on the PGI-I.
IPSS, International Prostate Symptom Score; MID, minimal important difference; OAB-q SF, Overactive Bladder Questionnaire short form; PGI-I, Patient Global Impression of Improvement.

**Table 3** Change scores for the IPSS and OAB-q SF by PGI-I outcomes: subgroup analysis for GP prescriptions

| PGI-I outcome | N (%) | IPSS | Missing | OAB-q SF | Missing |
|---|---|---|---|---|---|
| Very much better | 4 (3.1) | 18.0 (1.7 to 34.3) | 1 | 30.8 (−0.9 to 62.4) | 0 |
| Much better | 39 (30.2) | 9.2 (7.0 to 11.5) | 2 | 19.9 (14.8 to 24.9) | 2 |
| **A little better** | **57 (44.2)** | **5.4 (4.0 to 6.7)** | **3** | **11.2 (7.0 to 15.4)** | **3** |
| No change | 27 (20.9) | 3.1 (0.5 to 5.6) | 0 | 3.3 (−3.4 to 9.9) | 3 |
| A little worse | 2 (1.6) | −8.5 (−11.6.5 to 99.5) | 0 | −16.5 (−353.2 to 320.2) | 0 |

Change in IPSS and OAB-q SF scores were estimated by comparing symptom scores between baseline and 6 weeks. Mean change and 95% CIs are presented. Outcomes are inverted so that positive changes reflect symptom improvement. The PGI-I category 'a little better' reflects the MID for both questionnaires. None of the participants scored 'much worsened' or 'very much worsened' on the PGI-I.
IPSS, International Prostate Symptom Score; MID, minimal important difference; OAB-q SF, Overactive Bladder Questionnaire short form; PGI-I, Patient Global Impression of Improvement.

that three methodological differences account for these differences.

First, Barry *et al* compared patients between baseline and follow-up after 13 weeks. By contrast, the follow-up period in the current study was only 6 weeks. Although this difference of 7 weeks may have affected the ability of patients to recall their prior health state accurately, the true impact of this remains unclear. Change scores may also have been influenced by the natural variation that occurs in symptom severity over time.

Second, in the research by Barry *et al*, a different global assessment of patient improvement was used. This included a 5-point scale with the options 'marked improvement', 'moderate improvement', 'slight improvement', 'no improvement' and 'worse' for which the exact question was not reported. In our study, we used a 7-point Likert-type scale that ranged from 'very much better' to 'very much worsened'. We considered that this difference probably had no more than a marginal impact given that the positive outcome categories were comparable in both studies. Notably, none of the participants in our study reported that the symptoms had 'very' or 'very much' worsened.

Third, we mainly included men from primary care, rather than men solely from secondary care. Although it is generally thought that men in primary care have fewer symptoms, our men tended to have more severe symptoms (IPSS >19) than in the study by Barry *et al* (45% vs 25%). Barry *et al* also reported that baseline severity had a major impact on the MID, but when we compare their stratified analysis with ours, we had higher mean change scores for each PGI-I category. This might be explained by the fact that we only included men who actually used an alpha-blocker. In contrast to this focused approach, Barry *et al* used data for all participants in a large, randomised, double-blind trial of four treatment strategies for male LUTS. In their study, a lower MID could therefore have resulted from the inclusion of patients receiving placebo, finasteride, terazosin or combination therapy given that the efficacy of alpha-blockers exceeds that of both placebo and finasteride. The use of blinding meant that men who used placebo or finasteride may have overestimated their subjective improvement, while alpha-blocker users may have underestimated their subjective improvement. Given that the IPSS objectively counts symptoms, the placebo and finasteride users would experience a smaller change in the IPSS whereas the active drug users would experience a larger effect. Although the actual impact of each intervention is unknown, researchers in other fields have made similar observations.[10 11 24]

**Table 4** Change scores for the IPSS by PGI-I outcomes

| PGI-I outcome | Moderate symptoms (n=88) | | | Severe symptoms (n=73) | |
|---|---|---|---|---|---|
| | N (%) | Change | Missing | N (%) | Change |
| Very much better | 3 (3.4) | 6.5 (−50.7 to 63.7) | 1 | 3 (4.1) | 18 (1.7 to 34.3) |
| Much better | 30 (34.1) | 5.6 (3.7 to 7.5) | 1 | 19 (26.0) | 13.5 (10.5 to 16.5) |
| **A little better** | 33 (37.5) | 3.2 (1.7 to 4.8) | 0 | 32 (43.8) | 7.1 (5.3 to 9.0) |
| No change | 19 (21.6) | 1.3 (−1.7 to 4.3) | 0 | 19 (26.0) | 4.9 (2.3 to 7.6) |
| A little worse | 3 (3.4) | −5.0 (−31.0 to 20.9) | 0 | 0 (0.0) | – |

These results are stratified by baseline symptom severity on the IPSS: moderate symptoms are scores of 8–19 and severe symptoms are scores of ≥20. Change in IPSSs were estimated by comparing symptom scores between baseline and 6 weeks. Mean change and 95% CIs are presented. Outcomes are inverted so that positive changes reflect symptom improvement. The PGI-I category 'a little better' reflects the MID for both questionnaires. None of the participants scored 'much worsened' or 'very much worsened' on the PGI-I.
IPSS, International Prostate Symptom Score; MID, minimal important difference; OAB-q SF, Overactive Bladder Questionnaire short form; PGI-I, Patient Global Impression of Improvement.

We were unable to find any prior estimates of the MID for the OAB-q SF in the literature. Our finding that 11.0 points (95% CI 7.1 to 14.9) indicates a clinically relevant change is therefore a novel and important finding, but one for which the reliability will need to be assessed in other studies. We recognise that alpha-blockers are not specifically indicated for the treatment of OAB, but we contend that there is a considerable overlap with LUTS unrelated to OAB. Indeed, guidelines suggest prescribing alpha-blockers for most men with LUTS who request active treatment. This is because these agents have a rapid onset of action, good efficacy, and low rate and severity of adverse events.[3–5] We recommend further study to determine the MID in men with specific symptoms of OAB treated with anticholinergics or beta-3 agonists.

Some limitations need to be considered when assessing our results. Notably, the sample size of this study was small, which resulted in very low numbers of men being included in the PGI-I category 'very much better'. For that category, the mean change scores for both questionnaires showed very wide CIs. The same holds for the categories linked to symptom worsening. In those categories, a discontinuation trial, in which men stop their treatment, may be more suitable for reliable estimates. The sample size might also explain why the CI of the MID estimate for the OAB-q SF included the SEM.

Another limitation is reflected by difficulties we encountered with some of the associations between the PGI-I and the two PROM questionnaires. For the IPSS, the follow-up IPSS and PGI-I scores correlated better than with IPSS change and PGI-I scores, suggesting that this rating only reflected the current status, which in turn, decreases confidence in the MID estimate. For the OAB-q SF, the correlation coefficient between the baseline OAB-q SF and PGI-I scores was opposite in magnitude to that for the follow-up OAB-q SF and PGI-I scores. With both questionnaires, the correlation coefficients for the change scores were lower than the threshold of 0.5 that we set a priori.[21] High correlation coefficients are preferred between the anchor and the change in PROM, though some researchers have suggested applying lower thresholds.[25] Still, even the high correlation coefficients are insufficient to confirm that the transition rating is in fact measuring change as opposed to current health status.[21] Unfortunately, Barry et al did not report the correlation coefficients between the IPSS and the anchor,[2] which is consistent with most other research for PROMs.[26] Given the suboptimal relationship between the PROM and the anchor, we must stress that the estimates obtained for the MID should be interpreted with caution and should be confirmed in future investigations with larger samples.

In conclusion, this study is the first to define MID values for two important PROMs used to evaluate the effectiveness of treatment for male LUTS in primary care. Given that many men are treated in primary care, MID values for this setting are particularly important to inform evidence-based decision-making and to facilitate interpretation of the IPSS and OAB-q SF. Moreover, we consider that this study emphasises the importance of the MID to individual patients in daily practice. We defined the MID based on the PGI-I outcome 'a little better' in the present study, but patients may expect 'much better' as an outcome when starting therapy. To date, most outcomes of alpha-blocker and other drug treatments for male LUTS have been expressed as the mean IPSS change scores. In the vast majority of studies,[3] difference in IPSS changes between active treatment and placebo have approached, but not exceeded, the previously reported MID of 3.1.[1 2] Applying a threshold for improvement of 5.2 points, as described in our study, may change the interpretation of those studies.

**Author affiliations**
[1]Department of General Practice and Elderly Care Medicine, University of Groningen, University Medical Center Groningen, Groningen, The Netherlands
[2]Groningen Research Institute for Asthma and COPD (GRIAC), University of Groningen, University Medical Center Groningen, Groningen, The Netherlands
[3]Clinical Epidemiology and Biostatistics, Faculty of Health Sciences, McMaster University, Hamilton, Ontario, Canada
[4]Urology, Isala Hospitals, Zwolle, The Netherlands

**Acknowledgements** We are grateful for the assistance of Tom Vermist in collecting the data. The authors thank all the patients and collaborating pharmacies for their participation, as well as Dr Robert Sykes (www.doctored.org.uk) for providing editorial services.

**Contributors** MHB and MGS initiated the study. Acquisition of the data was done by MHB and MR. Analysis and interpretation of the data were done by MHB, HJA, TSD and HvdW. MHB wrote the manuscript with extensive support from HJA and TSD. All authors critically reviewed the manuscript.

**Funding** The Hein Hogerzeil Foundation supported this work with an unrestricted grant. The Foundation was not involved in the design of this study, nor in the data collection, data analyses, interpretation of the outcomes, or the writing of this manuscript.

**Competing interests** None declared.

**Patient consent for publication** Not required.

**Ethics approval** The medical ethics committee of the University Medical Centre Groningen, the Netherlands, approved the study (number 2016.122).

**Provenance and peer review** Not commissioned; externally peer reviewed.

**Data availability statement** Data are available upon reasonable request.

**ORCID iDs**
Marco H Blanker http://orcid.org/0000-0002-1086-8730
Tahira Sakina Devji http://orcid.org/0000-0001-8414-7410
Henk van der Worp http://orcid.org/0000-0001-5545-4155

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
