## [Reviewer comments · BMJ Open]

ARTICLE DETAILS

TITLE (PROVISIONAL)	Determining the minimal important differences in the International Prostate Symptom Score and Overactive Bladder Questionnaire: results from an observational cohort study in Dutch primary care
AUTHORS	Blanker, Marco; Alma, Harma; Devji, Tahira; Roelofs, Marjan; Steffens, Martijn; van der Worp, Henk

VERSION 1 – REVIEW

REVIEWER	Elaine McColl Newcastle University, UK
REVIEW RETURNED	29-Jul-2019

GENERAL COMMENTS	In general, this is an interesting paper, and a useful addition to the relevant literature. However, the following points needs consideration and addressing. Page 4, line 18: How can you be sure that '...the IPSS and OAB-q SF capture the spectrum of outcomes that are important to patients...' Page 4, lines 24-44: Greater clarity is needed here on precisely when and in what circumstances a primary care specific MID for IPSS and OAB-q SF might be used - for example, would it be when patients are recruited from primary care settings, when the intervention is delivered solely in primary care or under some other circumstances - in preference to the previously established MID for the IPSS. It is accepted that most men with LUTS first present to their GP, at least in the UK and the Netherlands, but many are subsequently referred to secondary care for further investigation and management. Page 5, line 31: Please justify the choice of 6 weeks as the follow-up interval, especially given that a longer interval was used in the previous study by Barry et al. Was true change expected over this shorter period. Page 5, line 40: Please add details of the response format for each item on IPSS. Page 6, line 36: Please justify why you used only the PGI-I category of 'a little better' as your anchor; others have used the absolute difference of change scores, and combined 'a little better' and 'a little worse' in calculating MIDs Page 8. line 16: It is of some concern that the mean mean change in IPSS for the PGI-I category of 'no change' was 3.1 points, with a 95% CI of 1.1-5.1 (overlapping with that for 'a little better', with a
---

	mean of 5.2, and that a similar pattern was observed for OAB-q SF. Was any consideration given to 'scaling' the values for 'a little better' and 'much better' by subtracting 3.1? [Incidentally, what was the mean for the 'no change' group in the previous study - reference 2?] Page 9, line 36. Do you really mean 'baseline' here, in your reference to Barry et al. It seems odd to have respondents carry out a global assessment of improvement (vis-a-vis what?) at baseline, rather than at follow-up. Page 10, line 60: it is unclear to me what is meant by 'trial discontinuation' in this particular context. Page 11, lines 10-38: You make a number of interesting observations here regarding the pattern of correlations between IPSS, OAB-q SF and PGI-I scores. Are you able to speculate on whether what you observe may represent a lack of test-retest reliability, or is it suggestive response shift?
--	---

REVIEWER	Emma Nyström Department of Public Health and Clinical Medicine, Unit of Research, Education and Development - Östersund, Umeå University, Sweden. My research group has organised workshops together with dr Marco Blanker and his research group on the topic of eHealth. We have however no common publications or research projects and I have no interests in the specific study of the article.
REVIEW RETURNED	05-Oct-2019

GENERAL COMMENTS	An interesting study in a clinically relevant population with a clearly defined intervention. MID provides information on the responsiveness of a specific PROM and offers a guide for interpretation. Many men with LUTS are treated with alpha-blockers in the primary care setting and the IPSS score is frequently used for evaluation, both in study settings and in everyday clinical work. To my knowledge this is the first study that establishes a MID in that population. Also looking at the OAB-q SF in this population provides new knowledge on improvement of which symptoms that are important when treating LUTS. However I have the following questions/concerns: Major revisions (general comments): MID are set to the mean of the group that expressed to be a little better. However the confidence intervals are overlapping between the groups expressing "no change" and "a little better", which calls upon carefulness when interpreting these MIDs. It is not clear from the manuscript what statistics were applied to analyse the differences in means between groups. Anyhow, as the aim was to define a minimum important difference I would consider the lack of significant difference a limitation that needs to be discussed. Perhaps it is true as the authors reason that in this patient group the expectation is rather to be "much better" from treatment but as the mean of the "a little better" was chosen as the MID, this deserves further discussion. It is also helpful to the reader when non-significant relationships are clearly shown in abstract, results section and tables. Furthermore on this topic, it is an interesting comparison to see that
--

	the men with different symptom severity had different MIDs. However with the small groups in the subgroup analyses confidence intervals are even wider and overlapping which makes it even more important to be careful with interpretation of these results and this is not clear from the abstract. Page 8, lines 20-25: As the authors correctly write in the methods section, the correlation analyses are performed to ensure that the IPSS and OAB-q SF measure the same underlying constructs. To me it makes no sense to correlate the PGI-I that expresses clinical change with single-point measurements. The reasons for performing these analyses would either need to be explained or omitted as they rather confuse the reader. Minor revisions (specific comments): Page 5, line 3: The PGI-I referred (ref 12) to is called the Patient Global Impression of Improvement, not the Perceived Global Impression of Improvement. This should also be corrected in line 26, as well as the tables 2, 3 and 4. Page 5, line 37: when referring to a tool as validated I find it useful to refer to the article(s) in which it was validated. Page 7, line 25: please clarify that this percentage was in the group that completed follow-up. page 8, line 21: Are you referring to the changes in score or the final follow-up score with the word "outcomes"? Page 9, line 5-14: As the study populations, the interventions and the global ratings are so different and the confidence intervals so wide I feel there is no point in comparing in this much detail. Table 1: I find the term "full participants" misleading as you report nothing about compliance. IQR - I find it more informative to report where the first quartile start and the third ends. tables 2-4: It could be clarified which groups were statistically different from each other. For the readability I also wonder whether it is necessary to abbreviate standard deviation and confidence interval within the text. If abbreviations are preferred, SD is more commonly abbreviated with capital letters, I believe also in BMJ open articles.
--	--

VERSION 1 – AUTHOR RESPONSE

Reviewer: 1

In general, this is an interesting paper, and a useful addition to the relevant literature. However, the following points need consideration and addressing.

Page 4, line 18: How can you be sure that '...the IPSS and OAB-q SF capture the spectrum of outcomes that are important to patients...'

>> Based on the development of both questionnaires, we made this statement. One can never be completely sure of course, but we have no reason to doubt this. For daily practice both questionnaires cover the issue of LUTS thoroughly. We have made no changes to the manuscript.

Page 4, lines 24-44: Greater clarity is needed here on precisely when and in what circumstances a primary care specific MID for IPSS and OAB-q SF might be used - for example, would it be when patients are recruited from primary care settings, when the intervention is delivered solely in primary care or under some other circumstances - in preference to the previously established MID for the IPSS. It is accepted that most men with LUTS first present to their GP, at least in the UK and the Netherlands, but many are subsequently referred to secondary care for further investigation and management.

>> MID may differ greatly between settings. (Our results illustrate that for the IPSS questionnaire.) Still, in guidelines, to date the MID for secondary care settings has been applied. As such this has resulted in assumptions that the threshold for a clinically relevant outcome is lower. As the majority of men with LUTS are treated in primary care, we feel that it is important to set clear treatment goals for that setting. Men need to be aware of what they can expect. Only when treatment is ineffective, a subsequent referral is made. We feel that this doesn't exclude the need to have proper MID values for the primary care setting. Unless we didn't catch the reviewers point here, we feel that we have explained this in the introduction section. We made no changes tot the manuscript.

Page 5,line 31: Please justify the choice of 6 weeks as the follow-up interval, especially given that a longer interval was used in the previous study by Barry et al. Was true change expected over this shorter period.

>> Alpha-blockers are short-acting drugs with clinical effect within a few weeks. As mentioned in the EAU guideline: Clinical effects take a few weeks to develop fully, but significant efficacy over placebo can occur within hours to days.

We have added this to the manuscript, with reference to the EAU guideline and additional reference. (Page 5): The period of six weeks was chosen as clinical effects of alpha-blockers take a few weeks to develop fully, but significant efficacy over placebo can occur within hours to days.

Page 5,line 40: Please add details of the response format for each item on IPSS.

>> We have added the following to this part: Each question has response options ranging from 0 to five, with higher scores reflecting more severe symptoms. Next, the full version of IPSS (and OAB-q-SF and PGI-I) have been added as supplementary files, which is also mentioned in the revised manuscript.

Page 6, line 36: Please justify why you used only the PGI-I category of 'a little better' as your anchor; others have used the absolute difference of change scores, and combined 'a little better' and 'a little worse' in calculating MIDs

>> By definition, the M of MID reflect Minimal important difference. So, the score reflecting the minimal effect has been chosen. This is considered standard procedure in this field. We don't see how a little better and a little worse should be combined when studying "minimal improvement" (which patients may be looking for). So, we have made no changes to the manuscript here.

Page 8. line 16: It is of some concern that the mean mean change in IPSS for the PGI-I category of 'no change' was 3.1 points, with a 95% CI of 1.1-5.1 (overlapping with that for 'a little better', with a mean of 5.2, and that a similar pattern was observed for OAB-q SF. Was any consideration given to 'scaling' the values for 'a little better' and 'much better' by subtracting 3.1?

>> We disagree with this approach, as the real threshold for minimal change is the observed change expressed by the little better group (as they did in fact feel improved). Below this threshold, observed changes could be random variations as patients do not really feel improved. Therefore, we did not consider scaling values in this way. We have chosen to present the data as is. We acknowledged the overlap between the outcomes, which is shown in many other studies on MID outcomes for other questionnaires, and could be explained by the relatively small sample in our study. We didn't change the manuscript.

[Incidentally, what was the mean for the 'no change' group in the previous study - reference 2?]

>> The 'no change' group in the previous study was the group of men who mentioned that they didn't experience any change in symptoms. We have added this explanation to the discussion.

In that study, the MID of 3.1 points (sd 0.27) fell within the 95% CI of the 'no change' group (**consisting of men who expressed that they hadn't experienced any change in symptoms**),

Page 9, line 36. Do you really mean 'baseline' here, in your reference to Barry et al. It seems odd to have respondents carry out a global assessment of improvement (vis-a-vis what?) at baseline, rather than at follow-up.

>> This is an error in our manuscript. We have corrected this. Second, in the research by Barry et al., a **different** global assessment of patient improvement was used.

Page 10, line 60: it is unclear to me what is meant by 'trial discontinuation' in this particular context.

>> We meant a discontinuation trial. We have changed this as follows: In those categories, a **discontinuation trial, in which men stop their treatment**, may be more suitable for reliable estimates.

Page 11, lines 10-38: You make a number of interesting observations here regarding the pattern of correlations between IPSS, OAB-q SF and PGI-I scores. Are you able to speculate on whether what you observe may represent a lack of test-retest reliability, or is it suggestive response shift?

>> In any follow-up using the same questionnaires, response shift may be present. The test-retest reliability of both questionnaires is high. We feel that we shouldn't speculate on this issue. We have added information on the test-retest reliability of both questionnaires to the method section. IPSS scores are often categorized as no/mild symptoms (0–7 points), moderate symptoms (8–19 points), or severe symptoms (≥ 20 points). **The IPSS was internally consistent (Cronbach's alpha = 0.86) and has excellent test-retest reliability (r = 0.92) [1].** The MID for the IPSS is currently considered to be 3.1 points. 2 The OAB-q SF contains six questions on 6-point Likert-type scales, with the outcomes transformed to a 0–100 point scale in which higher scores indicate more severe symptoms. 8 **This scale demonstrated good convergent validity, discriminant validity, internal reliability, reproducibility, and responsiveness to change [8].**

Reviewer: 2

An interesting study in a clinically relevant population with a clearly defined intervention. MIDs provides information on the responsiveness of a specific PROM and offers a guide for interpretation. Many men with LUTS are treated with alpha-blockers in the primary care setting and the IPSS score is frequently used for evaluation, both in study settings and in everyday clinical work. To my knowledge this is the first study that establishes a MID in that population. Also looking at the OAB-q SF in this population provides new knowledge on improvement of which symptoms that are important when treating LUTS. However I have the following questions/concerns:

Major revisions (general comments):

MID are set to the mean of the group that expressed to be a little better. However the confidence intervals are overlapping between the groups expressing "no change" and "a little better", which calls upon carefulness when interpreting these MIDs. It is not clear from the manuscript what statistics were applied to analyse the differences in means between groups. Anyhow, as the aim was to define a minimum important difference I would consider the lack of significant difference a limitation that needs to be discussed. Perhaps it is true as the authors reason that in this patient group the expectation is rather to be "much better" from treatment but as the mean of the "a little better" was chosen as the MID, this deserves further discussion. It is also helpful to the reader when non-significant relationships are clearly shown in abstract, results section and tables.

>> We thank the reviewer for this comment. We felt that there is no need to apply statistics to the mean values for the different PGI-I-outcomes, as this is captured in presenting the confidence intervals. As mentioned in the discussion section, we acknowledge that there is a small overlap between these two groups. Notably, in guidelines on male LUTS only the mean value of the IPSS-change score is mentioned, without any reference to the confidence interval shown in the study of Barry et al.

We have highlighted this throughout the manuscript. We had already discussed this in the first part of

the discussion.

Changes made to the manuscript:

Abstract (added to Results): For both questionnaires, confidence intervals showed an overlap with the no-change categories.

Strengths and limitations (added): Notably, the sample size of this study was small, which resulted in very low numbers of men being included in the PGI-I category 'very much better' or 'worsening of symptoms', **and may clarify the small overlap of the confidence intervals with the no-change category.**

Results (added): For both questionnaires, the confidence intervals of the MID-categories showed an overlap with the 'no change' categories.

Furthermore on this topic, it is an interesting comparison to see that the men with different symptom severity had different MIDs. However with the small groups in the subgroup analyses confidence intervals are even wider and overlapping which makes it even more important to be careful with interpretation of these results and this is not clear from the abstract.

>> We feel that the lack of precision, due to the small groups in the subgroup analyses is illustrated in the wider confidence intervals. As such, word count limits further explanation in the abstract. We have not changed the manuscript.

Page 8, lines 20-25: As the authors correctly write in the methods section, the correlation analyses are performed to ensure that the IPSS and OAB-q SF measure the same underlying constructs. To me it makes no sense to correlate the PGI-I that expresses clinical change with single-point measurements. The reasons for performing these analyses would either need to be explained or omitted as they rather confuse the reader.

>> We understand that this is a difficult point, and may confuse the readers. Correlations between questionnaire change scores and the anchor PGI-I should be obviously strong, as else these measure different concepts (e.g. a patient has improved on the questionnaire, but states that he/she feels much worse, does not make sense). Correlations between the anchor PGI-I and the baseline/follow-up questionnaire scores are performed to check for a possible response shift. Mostly anchor PGI-I scores seem correlated with follow-up scores (due to response shift). However this is not what you want. Therefore these correlations are checked also.

We have tried to explain the reasons for this in the methods section with reference to three full papers on this topic. We have added a few lines (in bold) to this:

The usefulness of anchor-based approaches depends on the relationship between the PROM and the anchor. 17-19 The anchor and PROM should be measuring the same or similar underlying constructs and should therefore be appreciably correlated. **Correlations between questionnaire change scores and the anchor PGI-I should be obviously strong, as else these measure different concepts. Correlations between the anchor PGI-I and the baseline and follow-up questionnaire scores are performed to check for a possible response shift. Mostly anchor PGI-I scores seem correlated with follow-up scores (due to response shift).** We therefore examined the Spearman correlation coefficients between the PGI-I and the IPSS and OAB-q SF for the baseline, follow-up and change data to ensure the anchor's validity. A correlation coefficient between the symptom change scores and the PGI-I of ≥ 0.50 , and an equal and opposite correlation of the PGI-I with the baseline score and the follow-up score, were considered ideal and likely to yield trustworthy MID estimates. 17-19).

Minor revisions (specific comments):

Page 5, line 3: The PGI-I referred (ref 12) to is called the Patient Global Impression of Improvement, not the Perceived Global Impression of Improvement. This should also be corrected in line 26, as well as the tables 2, 3 and 4.

>> We have corrected this throughout the manuscript.

Page 5, line 37: when referring to a tool as validated I find it useful to refer to the article(s) in which it was validated.

>> We have added two reference to the PGI-I:

13 Viktrup L, Hayes RP, Wang P, et al. Construct validation of patient global impression of severity (PGI-S) and improvement (PGI-I) questionnaires in the treatment of men with lower urinary tract symptoms secondary to benign prostatic hyperplasia. BMC Urol 2012;12:30,2490-12-30 doi:10.1186/1471-2490-12-30 [doi].

14 Yalcin I, Bump RC. Validation of two global impression questionnaires for incontinence. Am J Obstet Gynecol 2003;189:98-101.

Page 7, line 25: please clarify that this percentage was in the group that completed follow-up.

>> We have changed this section as follows: Notably, **86.3% of the participants** received their prescription from a GP and the remainder received it from a urologist.

page 8, line 21: Are you referring to the changes in score or the final follow-up score with the word "outcomes"?

>> With outcomes we refer to the association between IPSS change and PGI-I scores.

Page 9, line 5-14: As the study populations, the interventions and the global ratings are so different and the confidence intervals so wide I feel there is no point in comparing in this much detail.

>> We disagree with the reviewer at this point, as we want to clarify the differences in outcomes between the earlier study and ours. We therefore feel that all these differences need to be mentioned.

Table 1: I find the term "full participants" misleading as you report nothing about compliance.

>> We have changed this term into **Participants with completed follow-up** to avoid confusion.

IQR - I find it more informative to report where the first quartile start and the third ends.

>> We have added the requested information to Table 1.

tables 2-4: It could be clarified which groups were statistically different from each other.

>> We feel that presenting the confidence intervals reflects the statistical differences. We think that there is no need to only focus on "statistically significant" outcomes. We have made no changes to the manuscript here.

For the readability I also wonder whether it is necessary to abbreviate standard deviation and confidence interval within the text. If abbreviations are preferred, SD is more commonly abbreviated with capital letters, I believe also in BMJ open articles.

>> We couldn't find strict rules for this, but feel that 95%CI is so commonly used that is easier to read than using the full term. We have changed sd to SD throughout the manuscript.

VERSION 2 – REVIEW

REVIEWER	Elaine McColl Newcastle University, United Kingdom
REVIEW RETURNED	09-Nov-2019

GENERAL COMMENTS	In the main, the authors have responded well to feedback on the previous draft. However, the following points still need addressing. Page 4: Feedback on the previous draft asked 'how can you be sure that "...the IPSS and OAB-q SF capture the spectrum of outcomes that are important to patients"'. In your response you say "Based on the development of both questionnaires, we made this statement" and you go on to indicate that no change was made. A little more detail, e.g. that the IPSS and OAB-q development process involved qualitative data collection from patients (if this was indeed the case), would inspire greater confidence amongst readers in this assertion. I
---

	would ask that you add a clause or sentence to this effect. Page 4 - feedback on the last version asked you to make a stronger case for a primary care specific MID. You do so quite well in your response, but have not altered the paper to provide this level of clarity. I would ask that you do so, making it clear that the primary care specific value would be used in evaluations of interventions delivered in that setting, prior to referral to secondary care. Page 6 - your explanation of why you focused solely on 'a little better' rather than looking at the absolute value of change in those whose global rating of change was 'a little worse' or 'a little better' is clear and acceptable. However, as the latter approach has commonly been used in other studies establishing MIDs, I reiterate that it would be advisable to explicit about your rationale in the paper itself. Both reviewer 1 and reviewer 2 previously referred to the overlap on CIs for the 'no change' and 'a little better' groups, with reviewer 1 asking if you gave any consideration to 'scaling' to allow for this. You have responded to the effect that you've not made any changes to the MS in respect of this feedback. However, if two reviewers queried this point, it is likely that other readers may do so also. At the very least, some further explicit discussion in the paper of your rationale, as per the remarks in your feedback letter, would be helpful.
--	---

VERSION 2 – AUTHOR RESPONSE

Reviewer 1.

Page 4: Feedback on the previous draft asked 'how can you be sure that "...the IPSS and OAB-q SF capture the spectrum of outcomes that are important to patients". In your response you say "Based on the development of both questionnaires, we made this statement" and you go on to indicate that no change was made. A little more detail, e.g. that the IPSS and OAB-q development process involved qualitative data collection from patients (if this was indeed the case), would inspire greater confidence amongst readers in this assertion. I would ask that you add a clause or sentence to this effect.

>> It seems that the reviewer feels that the symptoms included in both questionnaires should be patient important. This is not the way we look at it. In the development of both questionnaires, researchers have tried to capture the symptoms that reflect a condition. For IPSS this was "benign prostatic hyperplasia" (a term that should not be used when considering symptoms), for OAB-q-SF this was overactive bladder syndrome. Both questionnaires capture these conditions, with some overlap. In the development of both questionnaires a quantitative approach was chosen, as is normal in such cases.

The patient importance comes with the level of change in both outcomes that are considered relevant by patients. This is reflected in the PGI-I.

We have changed the methods section as follows (page 5 and 6): **The IPSS questionnaire was originally validated as the American Urological Association Symptom Index for benign prostatic hyperplasia.¹ It includes 7 questions covering frequency, nocturia, weak urinary stream, hesitancy, intermittence, incomplete emptying and urgency.** Each question has response options ranging from 0 to five, with higher scores reflecting more severe symptoms. Total scores that may range from 0 (no symptoms) to 35 points (maximum score), and scores are often categorized as no/mild symptoms (0–7 points), moderate symptoms (8–19 points), or severe symptoms (≥20 points). The questionnaire was internally consistent (Cronbach's alpha = 0.86) and

has excellent test-retest reliability ($r = 0,92$).¹ The MID for the IPSS is currently considered to be 3.1 points.² **The AUA-SI has been internationally adopted and implemented worldwide under the namePSS.**

IPSS focuses on the concept of “benign prostatic hyperplasia” as cause of male LUTS, which appeared to have a multifactorial origin. Overactive bladder (OAB) is one of the alternative explanations of LUTS. Although urgency (included in the IPSS) relates to OAB, OAB includes other symptoms as well, which are not included in the IPSS questionnaire. Therefore, Coyne et al developed a condition specific questionnaire, the OAB-q.^{6,7} The OAB-q was developed from focus groups of men and women, clinician opinion, and a thorough literature review. More recently, this OAB-q has been shortened to benefit patients, researchers and clinicians.⁸ The OAB-q SF contains six questions on 6-point Likert-type scales, with the outcomes transformed to a 0–100 point scale in which higher scores indicate more severe symptoms.⁸ This scale demonstrated good convergent validity, discriminant validity, internal reliability, reproducibility, and responsiveness to change.⁸

Both IPSS and OAB-q-SF capture symptoms that are not by definition patient important, but rather reflect the conditions under study. To study if changes on a questionnaire over time are relevant for patients, the PGI-I has been developed using a quantitative approach.^{13, 15}

Page 4 - feedback on the last version asked you to make a stronger case for a primary care specific MID. You do so quite well in your response, but have not altered the paper to provide this level of clarity. I would ask that you do so, making it clear that the primary care specific value would be used in evaluations of interventions delivered in that setting, prior to referral to secondary care.

>> we have added the following to the introduction section (page 4): **To date the MID for secondary care settings has been applied in guidelines for primary care. 3, 5 It is unclear if applying the threshold for a clinically relevant outcome is appropriate. Men who receive treatment need to be aware of what can be expected.**

Page 6 - your explanation of why you focused solely on 'a little better' rather than looking at the absolute value of change in those whose global rating of change was 'a little worse' or 'a little better' is clear and acceptable. However, as the latter approach has commonly been used in other studies establishing MIDs, I reiterate that it would be advisable to explicit about your rationale in the paper itself.

>> We have added a sentence on this as follows (page 7): We defined the MID as the mean change in IPSS or OAB-q SF for the PGI-I category 'a little better', **as the M in MID reflects the *minimal* change that is considered relevant.** We also present the mean change scores for the other PGI-I categories.

Both reviewer 1 and reviewer 2 previously referred to the overlap on CIs for the 'no change' and 'a little better' groups, with reviewer 1 asking if you gave any consideration to 'scaling' to allow for this. You have responded to the effect that you've not made any changes to the MS in respect of this feedback. However, if two reviewers queried this point, it is likely that other readers may do so also. At the very least, some further explicit discussion in the paper of your rationale, as per the remarks in your feedback letter, would be helpful.

>> The overlap in outcomes has been mentioned in the abstract (both results and conclusion), limitations of the study, results section and the discussion paragraph. In part, this was included in the original manuscript; in part this was in response to the reviewers feedback. We therefore are surprised by this comment mentioning that we did not change the manuscript in response to the comments made.

We have now added the following for more clarity. I hope you feel that this is sufficient.

Page 12: In the current study, there was also some overlap between the CIs of the 'no change' and the 'a little better' group, though this was within a change of only 3.9 to 5.1 points. **This could be explained by the relative small samples in the subgroup analyses.**

VERSION 3 – REVIEW

REVIEWER	Elaine McColl Population Health Sciences Institute Newcastle University United Kingdom
REVIEW RETURNED	01-Dec-2019
GENERAL COMMENTS	I am content with the latest revisions.